# The Influence of Caerulomycin A on the Intestinal Microbiota in SD Rats

**DOI:** 10.3390/md18050277

**Published:** 2020-05-22

**Authors:** Hongwei Zhang, Mengmeng Lan, Guodong Cui, Weiming Zhu

**Affiliations:** 1Key Laboratory of Marine Drugs, Ministry of Education of China, School of Medicine and Pharmacy, Ocean University of China, Qingdao 266003, China; zhwhhxx@sina.com (H.Z.); lanmengmeng90@163.com (M.L.); cgd19940721@163.com (G.C.); 2Open Studio for Druggability Research of Marine Natural Products, Laboratory for Marine Drugs and Bioproducts, Pilot National Laboratory for Marine Science and Technology (Qingdao), Qingdao 266003, China

**Keywords:** intestinal flora, 16S rDNA gene high-throughput sequencing, caerulomycin A, colorectal cancer

## Abstract

Caerulomycin A (CRM A) is the first example of natural caerulomycins with a 2,2′-bipyridyl ring core and 6-aldoxime functional group from *Streptomyces caeruleus* and recently from marine-derived *Actinoalloteichus cyanogriseus* WH1-2216-6. Our previous study revealed that CRM A showed anti-tumor activity against human colorectal cancer (CRC) both in vitro and in vivo. Because some intestinal flora can affect the occurrence and development of CRC, the influence of CRM A on the intestinal flora is worthy of study in Sprague–Dawley (SD) rats. The high throughput sequencing of the V3-V4 hypervariable region in bacterial 16S rDNA gene results showed that the CRM A affected the diversity of intestinal flora of the SD rats treated with CRM A for 2, 3 and 4 weeks. Further analysis indicated that the abundance of genera *Prevotella_1*, *Prevotellaceae_UCG-001*, and *Lactobacillus* were increased while the that of genera *Alloprevotella* and *Ruminiclostridium_1* were decreased. For the CRC related intestinal flora, the abundance of genera *Bacteroides*, *Fusobacterium*, *Enterococcus*, *Escherichia-Shigella*, *Klebsiella*, *Streptococcus*, *Ruminococcus_2*, and *Peptococcus* of SD rats treated with CRM A were decreased, while that of abundance of genera *Bifidobacterium*, *Lactobacillus*, *Faecalibacterium*, *Blautia*, *Oscillibacter*, and *Clostridium* were increased. The results indicated that CRM A could influence the intestinal flora by inhibiting some species of harmful flora and improving the beneficial bacteria in intestinal flora in the SD rats. The results may provide a new idea for revealing the mechanism of the anti-CRC activity of CRM A.

## 1. Introduction

Caerulomycin A (CRM A) is a member of natural products containing a 2,2′-bipyridyl core structure, first isolated from the *Streptomyces caeruleus* by Funk and Divei in 1959 [1]. We also isolated it from the marine-derived *Actinoalloteichus cyanogriseus* WH1-2216-6 in 2011 [2] and demonstrated its biosynthesis pathway in the later cooperative research [3,4,5,6], which afforded the possibility for large yield and druggability study of CRM A. CRM A has been shown to exhibit diverse bioactivities such as antibacterial [2], immunosuppression [7,8], and antitumor activity against colorectal cancer (CRC) both in vitro and in vivo [2,9,10,11]. CRC is one of the most common malignancies worldwide, the third-most commonly-diagnosed cancer, and the second most common cause of death in cancer globally [12]. More and more evidence indicates that the occurrence and development of CRC is closely related to the imbalance of human intestinal microbiota. The intestinal microbiota of CRC patients is distinctly different from those in healthy people [13,14]. Imbalance of human gut microbiota might result in CRC [15]. For example, *Firmicutes* and *Bacterioidetes* are two major microbiota groups, accounting for about 90% of the gut microbiota in healthy people [16], while *Deffibacteres*, *Fusobacteria*, *Proteobacteria*, *Tenericutes*, and *Verrucomicrobia* are less abundant [17]. However, CRC patients had higher levels of *Fusobacterium*, *Escherichia*, and *Peptostreptococcus*, among which *Fusobacterium nucleatum*, *Escherichia coli*, and *Peptostreptococcus stomatis* were highly expressed in CRC patients [14,15]. To better understand the anti-CRC effect of CRM A, we studied the influence of CRM A on the gut microbiota in Sprague–Dawley (SD) rats. The flow chart of this study is shown in Figure 1.

## 2. Results

### 2.1. Sequencing Coverage of the Objects of Study

A total of 19,846~51,218 clean tags were generated from 40 samples which divided into four groups (Appendix A). The number of valid tags in this study was between 17,347 and 46,102 and the average length of the valid tags varied from 427.37 bp to 434.0 bp. All sequences could be clustered into 871 to 1,292 of operational taxonomic units (OTUs) and each OTU had shared 97% sequence identity. The total number of OTUs was 3,980 from 40 samples of all the groups.

### 2.2. Dilution Curves of Microorganisms in Samples

The dilution curve was used to determine whether the sequencing depth was sufficient to cover all microbial species and indirectly reflect the species richness in the samples. When the curve reached the plateau stage, it could be concluded that the sequencing depth had covered all species in the samples. As shown in Figure 2, the inflection point appeared around a sequence number of 5000 with the sequencing depth increasing. When the sequencing quantity of each sample exceeded 10,000, the curves tended to be flat and stopped rising, indicating that no new microorganisms would be amplified even if the sequencing quantity was increased. It was conclusion that the data of sample sequencing in each group were reasonable.

### 2.3. Alpha Diversity of the Intestinal Flora

The microbial diversity analysis showed that administration of CRM A could influence the whole microbial diversity in the colon. As shown in Figure 3, Alpha diversity analysis of the intestinal samples was performed based on the OTUs. Compared with the control, the microbial diversity of 2-week group and 4-week group increased slightly, but there was no significant difference between them (*p* > 0.05). However, the diversity was remarkably reduced after administration of CRM A for 3 weeks according to the analysis of the Chao-1 index, goods_coverage index, observed_species index, and Simpson index (*p* < 0.05). Compared to the 2-week group and 4-week group, the microbial diversity of the 3-week group was also significantly reduced (*p* < 0.05). The observed_species index and Chao1 index were used to calculate the abundance of bacteria; Shannon index, Simpson index, and PD_whole_tree index were used to calculate the diversity of bacteria. Therefore, the abundance and diversity of the intestinal flora in the colon of SD rats could be affected by CRM A after 3 weeks of administration.

### 2.4. Beta Diversity of Intestinal Flora

In order to understand the effects of CRM A on the intestinal microflora profile of SD rats, principal coordinate analysis (PCoA) of Bray Curtis distance was performed based on OTU. The results of PCoA showed the species diversity among the samples. The closer the samples were, the more similar the composition of the microbial species were. As shown in Figure 4, the percentage of the first principal component in 3-week group and 4-week group were obviously higher than that in the 0-week group. This indicated that the intestinal flora of SD rats treated by CRM A for 3 and 4 weeks were significantly different from that of the 0-week group and the 2-week group. Additionally, the percentage of the second principal component in 3-week group was obviously lower than that of other groups. It demonstrated that intestinal microbial communities of the SD rats administrated by CRM A for 3 weeks obviously changed. Results of both alpha and beta diversity analyses indicated that CRM A could affect the intestinal flora. Particularly, it may decrease the diversity of bacterial in the colon in SD rats after administration of CRM A for 3 weeks.

### 2.5. LEfSe Analysis

Linear discriminant analysis Effect Size (LDA Effect Size, LEfSe) analysis was conducted to estimate the impact of the abundance of each component and to identify the communities or species that had significant difference effect on the division of samples by LDA. The LDA score (log10) more than 3 was applied to identify the communities or species that have significant difference effect on CRM A. The result was shown in Figure 5. According to the LDA score, 33 taxa showed a significant difference in abundance between the CRM A treated and control groups (LDA score log10 > 4). Compared with the control group, the significant difference (LDA score log10 > 4) in intestinal microflora was found as the following: *Muribaculaceae* which belonging to the *Bacteroidales* (2-week group); *Prevotella_1*, *Prevotella_9*, and *Desulfovibrionaceae* (3-week group); *Firmicute*, *Lactobacillus*, *Lactobacillaceae*, and *Bacilli* (4-week group).

### 2.6. The Influence of CRM A on the Structure of Intestinal Flora of SD Rats

The histograms of species profiling were generated according to the species annotation results. A histogram of the relative abundance of species can be used to identify species with a higher relative abundance in each group and individual sample at different classification levels. The structure of intestinal flora at the level of phylum and genus were shown in Figure 6. In the colon, the top six abundant phyla of flora were *Bacteroidetes*, *Firmicutes*, *Proteobacteria*, *Epsilonbacteraeota Tenericutes*, and *Actinobacteria* which had accounted for over 99.5% of the total bacteria (shown in Table 1 and Figure 6A). The ratio of the *Bacteroidetes* in SD rats was decreased when treated with CRM A for 3 and 4 weeks while increased for 2 weeks compared with the control. The change trend of *Firmicutes* was opposite. At the level of genus, as shown in Figure 6B, *Prevotella_9*, *Prevotellaceae_UCG-003*, *Lactobacillus*, *Lachnospiraceae_NK4A136_group*, *Alloprevotella*, *Escherichia-Shigella*, *Bacteroides*, *Prevotella_1*, *Ruminococcus_1*, and *Ruminococcaceae_UCG-014* were the top 10 abundant bacteria which accounted for around 45% (Table 2). Proportion of the *Prevotella_9* and *Lactobacillus* were both remarkably increased (*p* < 0.001), while that of *Prevotellaceae_UCG-003* and *Alloprevotella* were decreased in SD rats treated with CRM A for 3 or 4 weeks. In addition, *Prevotella_1* was especially increased in 3-week group.

### 2.7. The Influence of CRM A on the Structure of Intestinal Flora of SD Rats Relate to CRC

The changes of intestinal flora related to CRC were also analyzed in this research. The relative floras at the level of phylum, family, and genus were shown in Table 3, Table 4 and Table 5 respectively. At the level of phylum (Table 3), the abundance of *Firmicutes* was slightly decreased in the 2-week group while significantly increased in the 3-week group and 4-week group compared with the control group (*p* < 0.05), which showed that the proportion of *Firmicutes* in the intestinal flora of SD rats increased by treated with CRM A for 3 or 4 weeks. The abundance of *Bacteroidetes*, *Proteobacteria,* and *Fusobacteria* were decreased in the 3-week group and 4-week group compared with the control group which showed that CRM A could inhibit *Bacteroidetes*, *Proteobacteria,* and *Fusobacteria* in SD rats for 3 or 4 weeks of administration, especially *Proteobacteria* and *Fusobacteria* (*p* < 0.05). According to the previous research [18,19], the abundance of *Bacteroidetes*, *Fusobacteria*, and *Proteobacteria* were increased, while *Firmicutes* was decreased in CRC patients.

At the level of family (Table 4), the abundance of *Bifidobacteriaceae*, *Lactobacillaceae*, and *Lachnospiraceae* were significantly increased by treating with CRM A for 3 or 4 weeks. Especially for the *Lactobacillaceae*, the percentage increased by 3% in the 4-week group compared with the control (*p* < 0.001). These microorganisms were confirmed to improve the symptoms colorectal cancer [20,21].

At the level of genus (Table 5), the overall trend of proportion of *Bacteroides*, *Fusobacterium*, *Enterococcus*, *Escherichia-Shigella*, *Escherichia*, *Klebsiella*, *Streptococcus*, *Ruminococcus_2*, and *Peptococcus* in the intestinal flora was declined by administrating with CRM A, especially for the *Fusobacterium* and *Streptococcus* (*p* < 0.05). The abundance of *Bacteroides* dropped by 0.95%, 1.06%, 1.44% by treating CRM A for 2, 3 and 4 weeks respectively. The ratio of *Bifidobacterium*, *Lactobacillus*, *Faecalibacterium*, *Blautia*, *Oscillibacter*, and *Clostridium* were increased, especially for the *Lactobacillus* (*p* < 0.001). Proportion of *Lactobacillus* reached highest after 4 weeks of administration, increasing by 3.08%. Another probiotic, *Bifidobacterium*, increased by 0.22% by treating with CRM A for 4 weeks. The abundance of *Bacteroides*, *Peptostreptococcu*, *Fusobacterium*, *Porphyromonas*, *Enterococcus*, *Escherichia-Shigella*, *Klebsiella*, and *Streptococcus* were increased and the abundance of *Bifidobacterium*, *Faecalibacterium*, *Oscillibacter*, and *Clostridium* were decreased in CRC people according to the previous studies [20,21,22].

## 3. Discussion

A growing number of studies have shown that intestinal flora can be directly or indirectly involved in the development of colorectal cancer. *Fusobacterium nucleatum*, *Bacteroides fragilis*, *Escherichia coli*, *Streptococcus gallolyticus*, and *Enterococcus faecalis* have been shown to promote the development of CRC while *Bifidobacterium*, *Lactobacillus*, *Faecalibacterium*, *Blautia*, *Oscillibacter*, and *Clostridium* are negatively associated with CRC [18,19,20,21,22]. Previous research confirmed that CRM A has significant activity of anti-CRC both in vitro and in vivo [2,9,10,11]. However, it is not clear whether it can modulate the intestinal flora or not. We studied the effects of CRM A on intestinal flora in SD rats and found CRM A could regulate the structure of intestinal flora and the abundance of a variety of intestinal microorganisms involved in CRC.

CRM A could affect the structure of the gut microbial community in SD rats, especially the microbiota related to the CRC. It could decrease the proportions of *Bacteroides*, *Fusobacterium*, *Enterococcus*, *Escherichia-Shigella*, *Klebsiella*, *Streptococcus*, *Ruminococcus_2*, and *Peptococcus* and increase those of *Bifidobacterium*, *Lactobacillus*, *Faecalibacterium*, *Blautia*, *Oscillibacter*, and *Clostridium*. According to the recent studies, among the intestinal microbiota positively related to CRC, *Fusobacterium nucleatum*, *Bacteroides fragilis*, and *Escherichia coli* are mostly investigated. *Fusobacterium nucleatum* is enriched in patients with CRC which is involved in regulating tumor immune-evasion processes by inhibiting the activity of natural killer (NK) cells targeting cancer cells [23,24]. It can also modulate the response to CRC therapy through inducing the activation of autophagy machinery and inhibiting the activation of the toll-like receptor 4 (TLR4) pathway [25]. *Bacteroides fragilis* is another gut microbe that has been shown to be enriched in patients with CRC. *Enterotoxigenic Bacteroides fragilis* (subgroup of *Bacteroides fragilis*) has been proved to produce the toxin fragylisin, which can cleave Ecadherin on colonocytes, affect epithelial permeability and cause intestinal inflammation [26]. *Escherichia coli* harboring pks islands in the intestinal of CRC people can produce colibactin, a genotoxic metabolite which can induce hepatocyte growth factor production and enhance tumor cell proliferation [27]. Some intestinal microbiota which increased by CRM A are associated with a lower risk of CRC. *Lachnospiraceae* and *Clostridium* are confirmed to negatively associate with CRC [21]. These bacteria can produce short chain fatty acids in the colon, which can help maintain epithelial health and homeostasis [28]. *Bifidobacterium animalis* and *Streptococcus thermophilus* are decreased in patients with CRC and the lactic acid produced by them may increase intraluminal acidity in the colon and inhibit amino acid degradation [29]. *Lactobacillus* has been confirmed to relieve symptoms associated with diarrhea in CRC patients [30,31]. *Lactobacillus* and *Bifidobacterium* was shown to prevent intestinal toxicity in cancer patients treated with both radiotherapy and chemotherapy [32]. CRM A may improve CRC by increasing such beneficial bacteria in intestinal flora.

CRM A could regulate intestinal flora, but the underlying mechanism is yet to be completely unraveled. Previous research indicated that CRM A had a significant inhibitory effect on bacteria such as *Puccinia graminis*, *Escherichia coli*, *Staphylococcus pyogenes. var. aureus*, *Aerobacter aerogenes*, and *Candida albicans* [1,2]. Some of these bacteria are also part of the intestinal flora, therefore the CRM A could directly regulate intestinal flora. Due to its obvious inhibitory effect on some enteric microorganisms, there was a decline in species diversity of the intestinal flora of rats administrated with CRM A for 3 weeks. However, such decline disappeared with the duration of administration.

As we can see from the results above, CRM A can regulate the structure of the intestinal flora. It can also regulate the abundance of intestinal microbes involved in the CRC. The results suggest that CRM A may induce the anti-CRC activity by regulating intestinal flora.

## 4. Experimental Section

### 4.1. Chemicals and Reagents

The standard of CRM A was homemade from *A*. *cyanogriseus* WH1-2216-6 [2] and the purity was higher than 99.01% (HPLC). Tween-80, PEG300, and formic acid were purchased from Sigma. Methanol and acetonitrile were purchased from Anaqua Chemicals Supply. DMSO and sodium chloride were purchased from China National Pharmaceutical Industry Corporation Ltd.

### 4.2. Animals and Administration of CRM A

SD rats (male, 200 ± 20 g) were maintained in a temperature-controlled room (22~23 °C), with humidity at 55%, and on a 12 h light/dark cycles under specific pathogen-free conditions. The animals were fed with a regular chow diet and free access to water. All the animal experiments were approved by animal ethics committee of Ocean University of China (Approval No. OUC-SMP-2019-05-01).

### 4.3. High-Throughput Sequencing of V3–V4 Region of Bacterial 16S rDNA Gene

#### 4.3.1. Sample Collection

According to the duration of CRM A administration, 40 SD rats were respectively divided into 4 groups, 0 (control), 2 weeks, 3 weeks, and 4 weeks groups. The SD rats were administrated the CRM A by intragastric administration at 5 mg/kg. The SD rats in every group were euthanized after administration of CRM A for the specific time and the colons were collected. The colons were dissected, and the contents were put into the sterile doffer tubes and frozen in liquid nitrogen for 30 s immediately and then stored at −80 °C for a long time. All the operations were performed under aseptic conditions.

#### 4.3.2. Genome DNA Extraction and Amplification

180–220 mg of intestinal content was put into 2 mL of centrifugal tube and all the operations were on ice. Then the intestinal microbial genome DNA was extracted by the QIAamp 96 PowerFecal QIAcube HT kit following the manufacturer’s instructions. The quality and quantity of DNA was verified with NanoDrop and agarose gel. Extracted DNA was diluted to a concentration of 1 ng/μL and stored at −20 °C until further processing.

The diluted DNA was used as template for PCR amplification of bacterial 16S rDNA genes with the barcoded primers and Takara Ex Taq (Takara). Primers of the PCR were as follows: the forward primers were 343F-5′-TACGGRAGGCAGCAG-3′ and the reverse primers were 798R-5′-AGGGTATCTA A TCCT-3′. Then the primers were applied to specifically amplify the V3-V4 hypervariable regions of the 16 S rDNA gene.

#### 4.3.3. Library Construction

Amplicon quality was visualized using gel electrophoresis, purified with AMPure XP beads (Agencourt), and amplified for another round of PCR. After purified with the AMPure XP beads again, the final amplicon was quantified using Qubit dsDNA assay kit. Equal amounts of purified amplicon were pooled for subsequent sequencing. Amplifier sequencing platform HiSeq 2500 PE250 was used for sequencing.

#### 4.3.4. Bioinformatic Analysis

Raw sequencing data were in FASTQ format. Paired-end reads were then preprocessed using Trimmomatic software [33] to detect and cut off ambiguous bases (N). It also cut off low quality sequences with average quality scores below 20 using a sliding window trimming approach. After trimming, paired-end reads were assembled using FLASH software [34]. Parameters of assembly were: 10 bp of minimal overlapping, 200 bp of maximum overlapping, and 20% of maximum mismatch rate. Sequences were performed further denoising as follows: reads with ambiguous, homologous sequences or below 200 bp were abandoned. Reads with 75% of bases above Q20 were retained. Then, reads with chimera were detected and removed. These two steps were achieved using QIIME software (version 1.8.0) [35]. Clean reads were subjected to primer sequences removal and clustering to generate OTUs using Vsearch software with 97% similarity cutoff [36]. The representative read of each OTU was selected using QIIME package. All representative reads were annotated and blasted against Silva database Version 123 (16s rDNA) using RDP classifier (confidence threshold was 70%) [37].

Besides, the alpha diversity was evaluated through calculating the observed species, Shannon Wiener, Simpson’s diversity indices, as well as the Chao1 index. Then, the beta diversity was evaluated through determining the similarity among the microbial communities within QIIME. In this study, both weighted and unweighted calculations were carried out and the principal coordinate analysis (PCoA) was conducted [38]. In the meantime, LEfSe [39] was performed to identify the differentially represented bacterial taxa between the two groups at genus or higher taxonomy levels. The analysis of variance (Kruskal_Wallis) was used to test for differences in overall community composition.

## 5. Conclusions

In this study, the influence of CRM A on the intestinal flora of SD rats was investigated by administration for different numbers of days. Through the high throughput sequencing of the V3–V4 hypervariable region in bacterial 16S rDNA gene, the diversity of the intestinal flora in the colon was analyzed. The results showed that CRM A could reduce the diversity of the intestinal flora for 3 weeks of treatment, which may due to its antimicrobial effects. CRM A could also affect the structure of the intestinal flora of SD rats. The microbiota involved in CRC were also analyzed. The CRM A not only increased the abundance of probiotics such as *Lactobacillus* and *Bifidobacterium* but also decreased the ratio of the bacteria which prompt the development of CRC like *Bacteroides*, *Fusobacterium*, *Enterococcus*, *Escherichia-Shigella*, *Klebsiella*, *Streptococcus*, *Ruminococcus_2*, and *Peptococcus* in the colon. Therefore, the regulation of intestinal flora by CRM A may be an intermediate step of the anti-CRC activity. However, for the reason that the constitute of the intestinal flora of rats is quite different from the human, the inference that CRM A may inhibit the CRC by regulating the intestinal flora need to be further verified.

## Figures and Tables

**Figure 1 marinedrugs-18-00277-f001:**
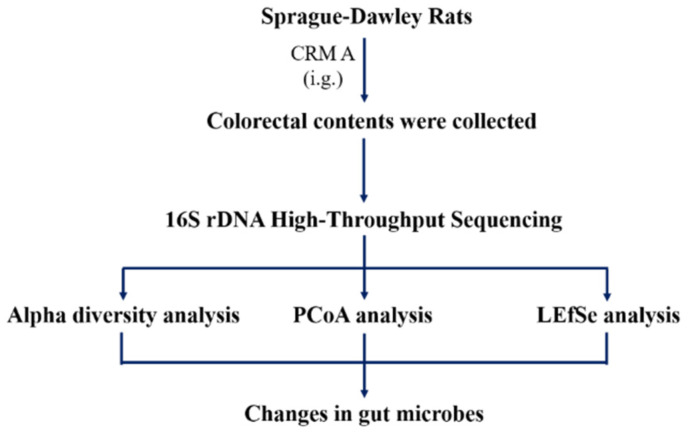
The flow chart of this study.

**Figure 2 marinedrugs-18-00277-f002:**
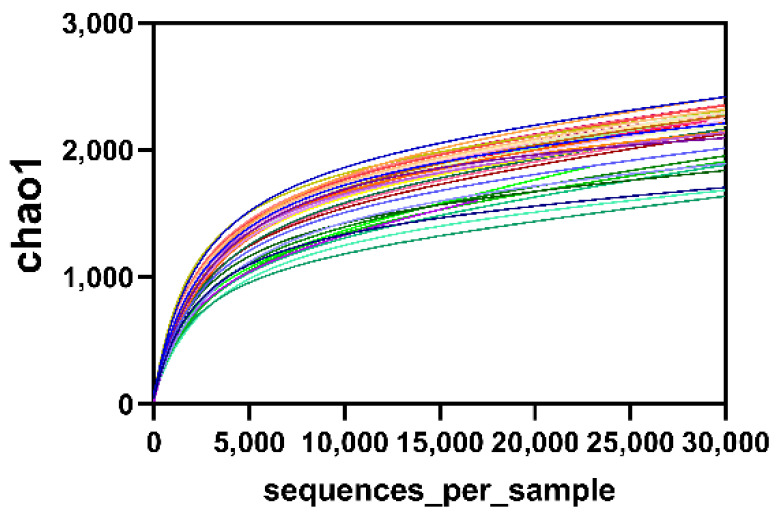
Rarefaction curves to the minimum sampling depth plotted for the Chao-1 index.

**Figure 3 marinedrugs-18-00277-f003:**
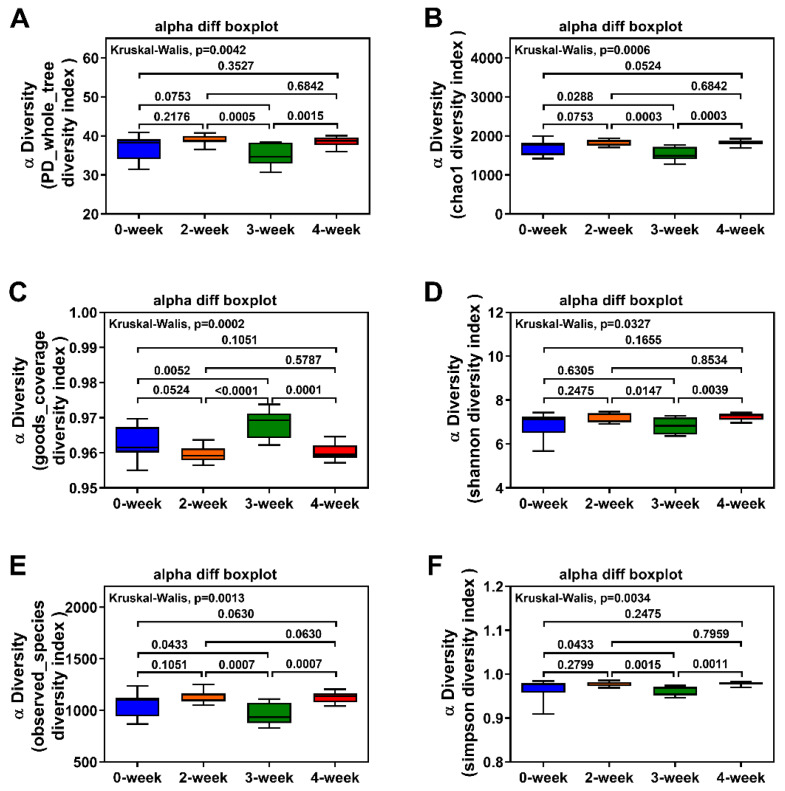
Alpha diversity of the intestinal samples. Histogram plotted for the PD _whole_ tree (**A**), Chao-1 index (**B**), goods_coverage (**C**), Shannon index (**D**), observed_ species (**E**), and Simpson index (**F**). 0-week, 2-week, 3-week, and 4-week mean the group which the SD rats administrated with the caerulomycin A (CRM A) for 0, 2, 3, and 4 weeks respectively.

**Figure 4 marinedrugs-18-00277-f004:**
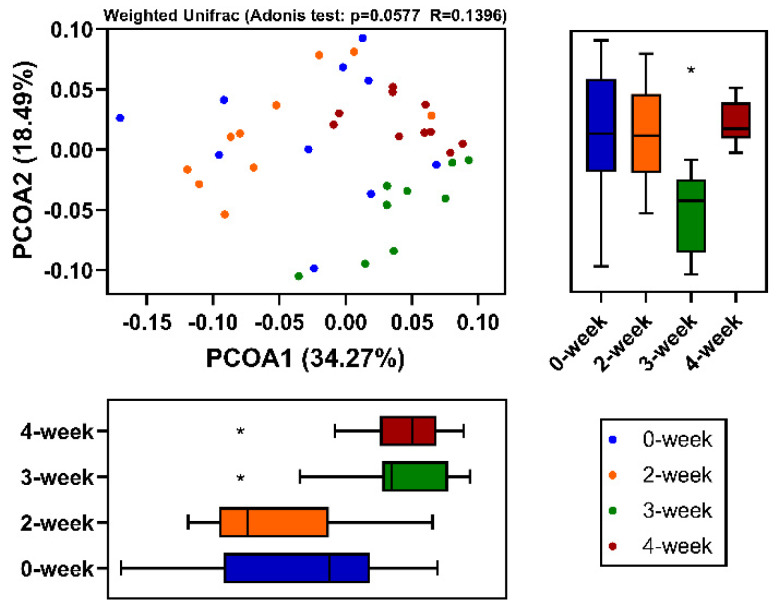
Principal coordinate analysis (PCoA) of 16S sequences using UniFrac distances. * *p* < 0.05.

**Figure 5 marinedrugs-18-00277-f005:**
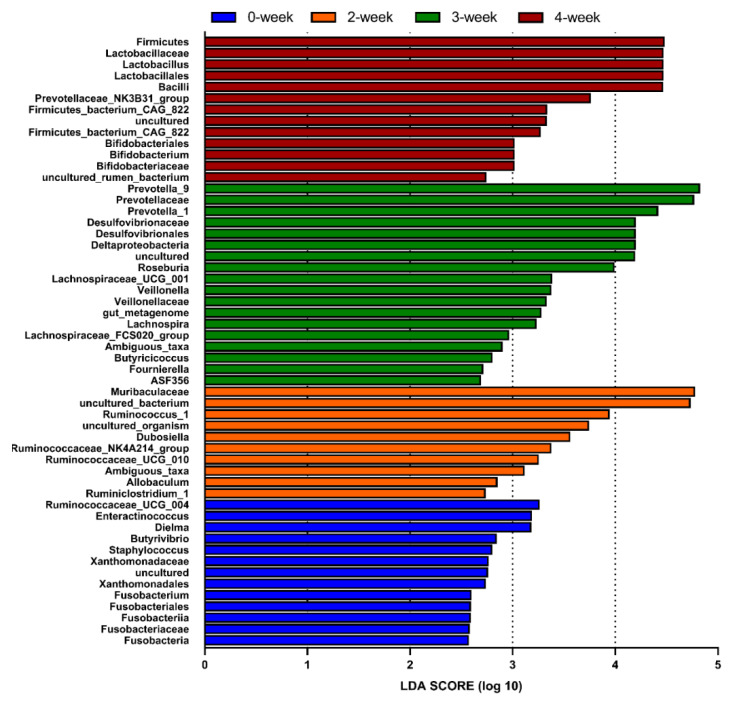
Difference in dominant microorganisms between groups in linear discriminant analysis (LDA) effect size (LefSe) analysis based on genus abundance. Control (0-week) and treated with CRM A (2-week, 3-week, and 4-week).

**Figure 6 marinedrugs-18-00277-f006:**
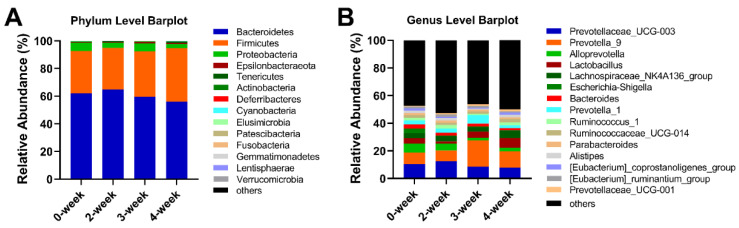
The histogram of community structure of intestinal flora in Sprague–Dawley (SD) rats at the level of phylum (**A**) and genus (**B**).

**Table 1 marinedrugs-18-00277-t001:** The relative abundance (%) and *p* value of the top six abundant bacteria of intestinal flora of SD rats at the phylum level.

Taxon	0-Week	2-Week	3-Week	4-Week	Test-Statistic	*p*
*Bacteroidetes*	62.1640	64.8152	59.5583	56.0251	3.3468	0.3412
*Firmicutes*	30.4237	30.1758	32.7556	38.8031	8.7585	0.0327
*Proteobacteria*	6.1519	3.7595	5.9350	2.8594	14.3512	0.0025
*Epsilonbacteraeota*	0.3358	0.2810	0.8496	0.8568	11.5744	0.0090
*Actinobacteria*	0.2681	0.3877	0.2443	0.3394	3.2626	0.3529
*Tenericutes*	0.3531	0.2191	0.2320	0.8129	11.2410	0.0105
Total	99.6966	99.6383	99.5748	99.6966		

**Table 2 marinedrugs-18-00277-t002:** The relative abundance (%) and *p* value of the top 10 abundant bacteria of intestinal flora of SD rats at the genus level.

Taxon	0-Week	2-Week	3-Week	4-Week	Test-Statistic	*p*
*Prevotellaceae_UCG-003*	10.2875	12.4177	8.4831	7.9037	5.4219	0.1434
*Prevotella_9*	8.6755	7.7603	19.1136	11.7028	17.2463	0.0006
*Alloprevotella*	6.1404	5.0177	1.8974	2.6281	21.7162	0.0001
*Lactobacillus*	4.0124	1.6408	4.3597	7.0909	17.1249	0.0007
*Lachnospiraceae_NK4A136_group*	3.7926	3.9663	3.3480	5.4652	7.4381	0.0592
*Escherichia-Shigella*	3.2305	0.1045	0.3733	0.0670	8.8984	0.0307
*Bacteroides*	3.1080	2.1532	2.0444	1.6718	1.9222	0.5887
*Prevotella_1*	2.7996	3.1260	6.1425	2.0343	14.4278	0.0024
*Ruminococcus_1*	1.8239	2.5618	1.0139	1.9687	16.5849	0.0009
*Ruminococcaceae_UCG-014*	1.8095	1.7453	1.6589	2.5085	4.9301	0.1770
Total	45.6799	40.4936	48.4348	43.0410		

**Table 3 marinedrugs-18-00277-t003:** The relative abundance (%) and *p* value of the bacteria of intestinal flora related to the colorectal cancer (CRC) at the phylum level.

Taxon	0-Week	2-Week	3-Week	4-Week	Test-Statistic	*p*
*Bacteroidetes*	62.1640	64.8152	59.5583	56.0251	3.3468	0.3412
*Firmicutes*	30.4237	30.1758	32.7556	38.8031	8.7585	0.0327
*Proteobacteria*	6.1519	3.7595	5.9350	2.8594	14.3512	0.0025
*Fusobacteria*	0.0173	0.0050	0.0000	0.0029	9.4593	0.0238

**Table 4 marinedrugs-18-00277-t004:** The relative abundance (%) and *p* value of the bacteria of intestinal flora related to the CRC at the family level.

Taxon	0-Week	2-Week	3-Week	4-Week	Test-Statistic	*p*
*Bifidobacteriaceae*	0.0418	0.1521	0.1319	0.2630	12.4351	0.0060
*Lactobacillaceae*	4.0124	1.6408	4.3597	7.0909	17.1249	0.0007
*Lachnospiraceae*	11.9536	11.7223	14.5629	16.2139	5.8054	0.1215

**Table 5 marinedrugs-18-00277-t005:** The relative abundance (%) and *p* value of the bacteria of intestinal flora related to the CRC at the genus level.

Taxon	0-Week	2-Week	3-Week	4-Week	Test-Statistic	*p*
*Bacteroides*	3.1080	2.1532	2.0444	1.6718	0.7965	0.5040
*Fusobacterium*	0.0173	0.0050	0.0000	0.0029	4.3240	0.0106
*Enterococcus*	0.0036	0.0022	0.0050	0.0007	0.9375	0.4326
*Escherichia-Shigella*	3.2305	0.1045	0.3733	0.0670	1.4653	0.2403
*Klebsiella*	0.0007	0.0036	0.0036	0.0000	1.5660	0.2145
*Streptococcus*	0.0072	0.0000	0.0014	0.0036	10.5590	0.0144
*Ruminococcus_2*	0.0454	0.0576	0.0295	0.0418	3.7683	0.2876
*Peptococcus*	0.0504	0.0447	0.0447	0.0368	0.6246	0.8908
*Bifidobacterium*	0.0418	0.1521	0.1319	0.2630	2.8433	0.0513
*Lactobacillus*	4.0124	1.6408	4.3597	7.0909	17.1249	0.0007
*Faecalibacterium*	0.0000	0.0036	0.0007	0.0014	2.4000	0.0838
*Blautia*	0.1175	0.1809	0.1081	0.1571	0.5695	0.6387
*Oscillibacter*	0.4749	0.4828	0.5693	0.4518	2.1415	0.5436
*Clostridium*	0.0699	0.0973	0.2212	0.2529	5.8599	0.1186

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
