# Peer review of "The Influence of Caerulomycin A on the Intestinal Microbiota in SD Rats"

_marinedrugs, 2020, doi:10.3390/md18050277_

Round 1

Reviewer 1 Report

The manuscript focuses on characterization of changes in the composition of mouse intestinal flora after treatment with Caerulomycin A (CRM A), a natural product with anti-colorectal cancer activity. The sequencing data from amplification of V3-V4 hypervariable regions of bacterial 16S rDNA gene was used to classify the intestinal flora upto 4 weeks of CRM A treatment.

Few minor comments are:

  1. Include a few statements on the beneficial aspect of the microbial flora that increases post-CRM A treatment.
  2. How accurate and unambiguous are the V3-V4 sequences in classifying microbial flora at genus and species phylogenetic level?
  3. Spell check required:e.g line 63,'reflecte"; page 168: "highst"
  4. Please expand the acronyms first time when they appear in the text . e.g OTU on line 60.

Author Response

Response to Reviewer 1 Comments

Point 1: Include a few statements on the beneficial aspect of the microbial flora that increases post-CRM A treatment.

Response 1: The statements have been added into the Discussion.

Point 2: How accurate and unambiguous are the V3-V4 sequences in classifying microbial flora at genus and species phylogenetic level?

Response 2: V3-V4 sequencing of Microbe is generally accurate to the genus level by default. The accuracy at the species level is limited due to the two aspects. One is the sequence length is limited and second is the number of species which the database annotates to is limited.

Point 3: Spell check required: e. g line 63,'reflecte"; page 168: "highst".

Response3: Spell check has been done throughout the manuscript.

Point 4: Please expand the acronyms first time when they appear in the text. e. g OTU on line 60.

Response 4: All the acronyms have expanded when they appeared first time.

Reviewer 2 Report

Proofread the entire document. Typos and grammar errors are detected.

Expand abbreviations when it is used for the first time in the document

Experimental design is not clear. Please include a flow chart

Line 225: Expand the section. How the animals were housed ? what was the diet?

Line 233: Whether the drug was administered orally or a intragastric injection

Line 60: Indicate the total number of OTUs from the total reads

Figure 3A: No clustering is observed. What was the stress threshold for pCoA analysis? Discuss the finding

LIne 211: Statement is vague and not connecting

Author Response

Response to Reviewer 2 Comments

Point 1: Proofread the entire document. Typos and grammar errors are detected. 

Response 1: Typos and grammar errors have been revised throughout the manuscript.

Point 2: Expand abbreviations when it is used for the first time in the document.

Response 2: All the acronyms have expanded when they appeared first time.

Point 3: Experimental design is not clear. Please include a flow chart.

Response 3: The flow chart has been drawn and added into the manuscript as Figure 1.

Point 4: Line 225: Expand the section. How the animals were housed? what was the diet? 

Response 4: All animals were maintained in a temperature-controlled room (22~23 â—¦C), with humidity at 55%, and on a 12-h light/dark cycles under specific pathogen-free conditions. The animals were fed with a regular chow diet and free access to water.

Point 5: Line 233: Whether the drug was administered orally or a intragastric injection?

Response 5: The drug was administered by intragastric injection.

Point 6: Line 60: Indicate the total number of OTUs from the total reads.

Response 6: The total number of OTUs was 3980 from the total reads, and the relevant content has been added into the manuscript.

Point 7: Figure 3A: No clustering is observed. What was the stress threshold for pCoA analysis? Discuss the finding.

Response 7: The clustering was not circled in this manuscript. However, the clustering effect among members of the same group was obvious, especially for the 3-week group and 4-week group. It is generally believed that stress threshold lower than 0.2 can be represented by NMDS two-dimensional point graph. Stress threshold is not generally used in PCoA analysis. According to the literature, PCoA analysis is usually showed as the P value in Adonis.

Point 8: Line 211: Statement is vague and not connecting.

Response 8: The statement has been revised in the manuscript after reconsideration.
